# Peer review of "High Levels of Admixture in Anopheles gambiae Populations from Côte d’Ivoire Revealed by Multilocus Genotyping"

_insects, 2022, doi:10.3390/insects13121090_

Round 1

Reviewer 1 Report

This relatively short research paper reports on recent findings of high interspecific hybridisation between the two most closely related species in the Anopheles gambiae complex, Anopheles gambiae and Anopheles coluzzii. The work presented advances that in a recent publication by the same authors by temporal extension of the samples from the two villages in Cote d’Ivoire and genotyping at an enlarged panel of multilocus ancestry informative markers, which are compared with the standard molecular species diagnostic on the X chromosome. Results show high levels of admixture and discordances between genome wide patterns and the species diagnostic marker, questioning the latter’s applicability in a way only previously encountered in ‘far west’ locations of extreme hybridisation and inter-specific mixture. Whilst the work perhaps raises more questions than it answers it gives additional insights into a biologically complex and consequential question, which should spur further work. The manuscript is well written and presented and I only have a few comments, which are minor.

Line 36 and 310 should read complicated

Line 52 in should be to

Line 57 delete high

Line 59-60 delete Despite and insert but before premating

Line 96 I think to hypothesize that should be to the hypothesis of

References 33 and 34 are the same

Line 111 and 278and 298 genotypization should be genotyping

Line 120 How many houses were sampled in each village?

Line 151 arose should be either arising or that arose

Line 175 Ancestral should read Ancestry

Author Response

Thank you very much for the positive comments! See below point by point answer to the reviewer 1 comments:

Line 36 and 310 should read complicated: Both of them have been rephrased ; see lines 35-37 and 323-327

Line 52 in should be to: done

Line 57 delete high: done

Line 59-60 delete Despite and insert but before premating: done

Line 96 I think to hypothesize that should be to the hypothesis of: done

References 33 and 34 are the same: this has been corrected

Line 111 and 278and 298 genotypization should be genotyping: done

Line 120 How many houses were sampled in each village?  The following sentence was added in lines 127-129_  “Mosquitoes were collected  during December 2018 and in March, May, and October 2019 by pyrethrum spray collections for four consecutive days/month in 5 randomly selected houses/ village”.

Line 151 arose should be either arising or that arose: changed into “that arose”

Line 175 Ancestral should read Ancestry: done

Reviewer 2 Report

General comments

The manuscript entitled "Genome-wide levels of admixture in Anopheles gambiae populations from Côte d'Ivoire revealed by multilocus genotyping" by Tondossama and colleagues reports unusually high frequencies of IGS and AIM-hybrids of Anopheles gambiae and Anopheles coluzzii in two populations of Côte d'Ivoire. They also observe incongruent results between IGS-diagnostic markers, chromosome-X AIMs, and chromosome-X and autosomal AIMs. 

Two villages were sampled, one was coastal (Ayame), and the other was inland (Petessou). Using AIM genotyping, 27% (8/29) were hybrid individuals in Ayame village. This percentage rose to 38% IGS-hybrid genotyping was performed. On the contrary, in the Petessou village, only 5% (1/19) were hybrid individuals using AIM markers and 7% when IGS-hybrid genotyping was performed. The high frequency of hybrids may be explained by a temporary breakdown of reproductive barriers due to increased inter-specific contact since An. gambiae and An. coluzzi occur in sympatry and the same abundance in this area. This unexpectedly high frequency also could be due to methodological biases, leading to underreporting of putative hybrids in other publications. The fact that no individuals represented by consistent heterozygous IGS and AIM patterns were observed allows disputing the current breakdown of reproductive barriers between An. coluzzii and An. gambiae hypothesis. If the hypothesis is true, most of the hybrids should be heterozygous, as expected for F1 hybrids. This result looks evident when Petessou village is considered since no AIM hybrids were found in the heterozygous IGS. The authors suggest that the observed patterns are due to cryptic taxa rather than hybridization between the two species. The authors recognize that samples from different years could help to confirm or dispute this hypothesis. 

I have some comments addressing both major and minor issues for the authors to consider:

Major Comments: 

One of my primary concerns is that the title does not reflect the entire manuscript (Simple Summary, abstract, discussion). Wide levels of admixture in Anopheles gambiae populations were indeed observed for AIM markers. However, the whole manuscript focuses on the high frequencies of IGS and AIM-hybrids in the Ayame and Petessou populations. I suggest reforming the title to reflect the excellent work done and its epidemiological and evolutionary relevance. 

Minor Comments: 

While the overall quality of the language is fine, there are some mistakes the authors should fix. Some (but not all) of which I have listed below:

Simple Summary

Line 15. "the transmission of" I suggest replacing it with "Transmitting."

Line 17. Please rewrite to clarify: "And" instead" as well as."

Line  18-19. Reorder to clarify: "consistently allow us to identify" instead of "to consistently identify."

Line 21. "due to the current" instead of "due to current" 

Line 23. "found" may be unnecessary in this sentence; consider removing it. 

Line 24. "allow an understanding of its potential" instead of "allow to understand its potential."

 Abstract

Line 28. except in restricted areas" instead of "with exception of restricted areas."

Line 30. "the existence of" may be unnecessary in this sentence; consider removing it.

Line 30. Remove the comma after "taxa."

Line 29. "the hypothesis" instead of "led to hypothesize that" 

Line 33. "The three chromosomes" instead of "the 3 chromosomes."

Line 36. "complicated pattern" instead of "complicate pattern."

Line 36. "that deserves deeper" instead of "which deserves deeper."

Line 37. "To understand better" instead of "to better understand."

Line 38. "existence of different cryptic taxa" instead of "existence of futher cryptic taxa."

Line 39. "in assessing" instead of "to assess."

Results

In Table 1, please consider adding a panel (on the right?) with the AIM results (numbers and their frequencies). 

In Figure 1, please consider adding a line to the rows; then, it will be easier to identify each individual in the panel. Additionally, the plot in the top right (2L) has an obliquus white line that disturbs the visualization. Please make this figure with better quality. 

In Table 2, Line 231 . "and" instead of "ant."

Author Response

Thank you very much for the positive comments!

Following your suggestion concerning the article title we changed it into a more general: “High levels of admixture in Anopheles gambiae populations from Côte d'Ivoire revealed by multilocus genotyping”

See below point by point answer to the reviewer 2 minor comments:

Line 15. "the transmission of" I suggest replacing it with "Transmitting." done

Line 17. Please rewrite to clarify: "And" instead" as well as." done

Line  18-19. Reorder to clarify: "consistently allow us to identify" instead of "to consistently identify." We prefer maintaining the previous phrasing and think changing it would slightly change the meaning of the sentence.

Line 21. "due to the current" instead of "due to current" done

Line 23. "found" may be unnecessary in this sentence; consider removing it. done

Line 24. "allow an understanding of its potential" instead of "allow to understand its potential." done

 Abstract

Line 28. except in restricted areas" instead of "with exception of restricted areas." done

Line 30. "the existence of" may be unnecessary in this sentence; consider removing it. done

Line 30. Remove the comma after "taxa." done

Line 29. "the hypothesis" instead of "led to hypothesize that" We prefer maintaining the previous phrasing

Line 33. "The three chromosomes" instead of "the 3 chromosomes." done

Line 36. "complicated pattern" instead of "complicate pattern." done

Line 36. "that deserves deeper" instead of "which deserves deeper." done

Line 37. "To understand better" instead of "to better understand." done

Line 38. "existence of different cryptic taxa" instead of "existence of futher cryptic taxa." done

Line 39. "in assessing" instead of "to assess." done

Results

In Table 1, please consider adding a panel (on the right?) with the AIM results (numbers and their frequencies). We did not add this panel since not all specimens included in table 1 were genotyped for AIMs and thus adding this panel could have been misleading; anyway all results concerning AIM genotyping are summarized in Table 2 ,Figure 1 and the supplementary material.

In Figure 1, please consider adding a line to the rows; then, it will be easier to identify each individual in the panel. Additionally, the plot in the top right (2L) has an obliquus white line that disturbs the visualization. Please make this figure with better quality. Thanks for this suggestion; Figure 1 was inserted now in the text in a different format with lines between each specimens; also a different file format can be made available to enhance quality.

In Table 2, Line 231 . "and" instead of "ant." ok